# Recognition of Chinese Electronic Medical Records for Rehabilitation Robots: Information Fusion Classification Strategy

**DOI:** 10.3390/s24175624

**Published:** 2024-08-30

**Authors:** Jiawei Chu, Xiu Kan, Yan Che, Wanqing Song, Kudreyko Aleksey, Zhengyuan Dong

**Affiliations:** 1School of Electronic and Electrical Engineering, Shanghai University of Engineering Science, Shanghai 201620, China; cjw5019@126.com (J.C.); xiu.kan@sues.edu.cn (X.K.); swqls@126.com (W.S.); 2Engineering Research Center of Big Data Application in Private Health Medicine, Fujian Province University, Putian 351100, China; 3New Engineering Industry College, Putian University, Putian 351100, China; 4Department of Medical Physics and Informatics, Bashkir State Medical University, Ufa 450008, Russia; akudreyko@bashgmu.ru; 5Department of General Physics, Ufa University of Science and Technology, Ufa 450076, Russia; 6School of Science, Donghua University, Shanghai 201620, China; dzydhu@163.com

**Keywords:** rehabilitation robots, named entity recognition, electronic medical records, deep learning, information fusion

## Abstract

Named entity recognition is a critical task in the electronic medical record management system for rehabilitation robots. Handwritten documents often contain spelling errors and illegible handwriting, and healthcare professionals frequently use different terminologies. These issues adversely affect the robot’s judgment and precise operations. Additionally, the same entity can have different meanings in various contexts, leading to category inconsistencies, which further increase the system’s complexity. To address these challenges, a novel medical entity recognition algorithm for Chinese electronic medical records is developed to enhance the processing and understanding capabilities of rehabilitation robots for patient data. This algorithm is based on a fusion classification strategy. Specifically, a preprocessing strategy is proposed according to clinical medical knowledge, which includes redefining entities, removing outliers, and eliminating invalid characters. Subsequently, a medical entity recognition model is developed to identify Chinese electronic medical records, thereby enhancing the data analysis capabilities of rehabilitation robots. To extract semantic information, the ALBERT network is utilized, and BILSTM and MHA networks are combined to capture the dependency relationships between words, overcoming the problem of different meanings for the same entity in different contexts. The CRF network is employed to determine the boundaries of different entities. The research results indicate that the proposed model significantly enhances the recognition accuracy of electronic medical texts by rehabilitation robots, particularly in accurately identifying entities and handling terminology diversity and contextual differences. This model effectively addresses the key challenges faced by rehabilitation robots in processing Chinese electronic medical texts, and holds important theoretical and practical value.

## 1. Introduction

Rehabilitation robots play a crucial role in modern medicine, including assisting with physical rehabilitation and providing remote medical support [1,2,3,4,5]. During rehabilitation, robots analyze patients’ electronic medical records to offer corresponding auxiliary medical plans, supporting doctors in decision-making and enhancing treatment effectiveness [6,7]. However, current named entity recognition technology faces challenges in handling Chinese electronic medical records. The complexity and diversity of Chinese medical terminology limits recognition accuracy and generalization capability, while variations in record structures and language styles further complicate algorithms [8,9,10,11]. Therefore, enhancing the application efficiency of rehabilitation robots in medical practice through advanced natural language processing and deep learning methods is crucial for the advancement of medicine.

Electronic medical records represent digital repositories of earlier versions of paper-based routine medical documentation, containing various types of semi-structured and unstructured text data. Accurate identification of classified medical entities from electronic medical records is crucial for subsequent research and applications. Particularly for rehabilitation robots, precise entity recognition can significantly enhance their decision-making and operational capabilities. In the past two decades, a series of research results have been achieved by applying natural language processing techniques for information recognition extraction from electronic medical records [12,13]. Most of these findings are based on the English electronic medical record annotated corpus dataset published by the Message Understanding Conference (MUC) and the United States Informatics for Integrating Biology and the Bedside(I2B2). However, there exist obvious differences between Chinese and English languages in methods of word formation, grammatical structures and sentence logic [14]. Making Chinese electronic medical records requires the development of linguistic characteristics by using neural networks; the aim of this study is to achieve this goal [15,16,17,18,19]. Current mainstream methods for named entity recognition research in Chinese electronic medical records fall into two categories: machine learning and deep learning. On the one hand, machine learning methods are mostly combined with rules to build named entity recognition models [20]. For instance, the patterns, rules and features of medical text corpora have been summarized and been integrated them with the CRF model to establish a statistical and rule-based Chinese medical institution name recognition model [21]. In order to adapt to the Chinese language characteristics, a multi-feature fusion approach has been introduced for Chinese electronic medical record entity recognition in [22]. Machine learning methods have significantly expanded the scope and accuracy of entity recognition, but they heavily rely on feature engineering, which can complicate feature selection, while ignoring cross-fertilization of medical domain knowledge [23,24].

On the other hand, deep learning methods mainly address the problem of named entity recognition in massive electronic medical records from the perspectives of reducing feature engineering dependency and improving the efficiency of using large amounts of data. For example, the attention mechanisms have been introduced into the LSTM model to recognize the electronic medical record entities [25,26,27]. The BERT-BiGRU-CRF model has been designed in [28], which incorporates the bidirectional Transformer structure from BERT to recognize contextual semantic relationships, addressing the limitation of traditional methods in representing polysemy. A multi-layer Transformer model has been established in [29], which employs multi-head and self-attention mechanisms to extract features from multiple semantic spaces of network security threat intelligence. However, it also brings challenges, such as the model being complex, the training time being long, the nested entities being difficult to extract, and the boundary of intersecting entities not being well recognized.

Based on the above analysis, Chinese electronic medical records are characterized by large volumes of data, high knowledge density, and complex vocabulary composition, which present significant challenges for rehabilitation robots in analyzing and understanding medical conditions. Current research in the field of named entity recognition has made significant progress, particularly in achieving high accuracy in handling simple entity recognition tasks. However, when dealing with complex and highly specialized texts, such as Chinese electronic medical records, certain limitations still exist: (1) Chinese electronic medical records primarily consist of declarative phrases, and although the forms of entities are relatively simple, their semantic structures are complex, which can easily lead to the loss of important information during recognition; (2) the models perform poorly when faced with the polysemy of the same entity in different contexts, thereby limiting the depth of understanding of medical records by rehabilitation robots; and (3) the cross-expression of different entities in medical records leads to unclear recognition of entity boundaries by rehabilitation robots. To address these issues and improve model accuracy, a novel named entity recognition method is proposed in this paper. The main contributions include: (1) although existing research performs well in recognizing simple entities, it still falls short in capturing complex semantic structures. Therefore, a more refined entity definition strategy and data preprocessing scheme are designed to optimize the practicality of data specific to Chinese electronic medical records, reducing the loss of information. (2) To address the poor performance of existing research in handling the polysemy of the same entity in different contexts, a lightweight named entity recognition model for electronic medical records is proposed, which uses knowledge fusion and multi-task joint training. This approach not only reduces computational burden, but also enhances the understanding of entity expansion and categorization, significantly improving the depth of medical record understanding by rehabilitation robots. And (3) although progress has been made in clearly defining the boundaries of single entities, the existing research still shows limitations when dealing with the cross-expression of different entities and fuzzy boundaries in electronic medical records. This study combines MHA and CRF to establish a new entity boundary recognition model, effectively improving the ability of rehabilitation robots to recognize complex entity boundaries and enhancing overall recognition accuracy.

The structure of the paper is as follows. Detailed description of the problem and data processing scheme are presented in Section 2. In Section 3, a common framework of development of knowledge fusion is shown in Section 2. A case study in Section 4 demonstrates ablation and comparative experiments, and shows the effectiveness and feasibility of the proposed model. Our conclusions are presented in Section 5.

## 2. Description of the Problem and Data Processing Scheme

### 2.1. Data Acquisition and Analysis

When reading electronic medical records, rehabilitation robots need to use NLP technology to transform unstructured text data into structured information. This requirement arises due to the frequent presence of incomplete sentence structures, numerous special symbols and abbreviations, and unclear entity categories and boundaries in Chinese electronic medical records. The process involves data preprocessing, word segmentation, entity recognition, and relationship extraction. Subsequently, the robots provide corresponding treatment recommendations based on the actual medical records of the patients. To ensure the practicality and accuracy of the experiment, the public dataset Yidu-S4K is used in this study. This dataset contains real medical data manually annotated by a professional medical team, and is used for the “Named Entity Recognition for Chinese Electronic Medical Records” task in the CCKS 2019 competition. The dataset embodies the typical characteristics of Chinese electronic medical records, including 1000 training samples and 379 test samples. Six types of entities are manually annotated in this dataset: disease and diagnosis, imaging examination, laboratory test, surgery, medication, and anatomical site. The number of each type of entity in the training and test sets is shown in Table 1.

Electronic medical records have distinct language structures and linguistic characteristics that set them apart from general text. Unlike identifying three major classes and seven sub-classes of the entities in general text, electronic medical records mainly involve the recognition of medical entities, e.g., symptoms, laboratory tests, and procedures. Representation of electronic medical record texts is unique, as symbols carry specific meanings that contribute to the increased complexity of entity recognition. For instance, in the context of “White blood cells level 3+”, the symbol “+” does not function as a typical mathematical addition sign found in general texts; instead, it signifies a significant increase in the quantity of white blood cells. Entities in electronic medical records often contain modifiers and everyday vocabulary, making it challenging to identify their left boundaries. For example, entities like “portal vein”, “inferior vena cava”, and “loose stool” contain everyday words that complicate recognition.

Meanwhile, there are typical phenomena in Chinese electronic medical records, which include: (1) Polysemy: Identifying multiple meanings of words. For instance, “severe regurgitation of mitral and tricuspid valves with increased regurgitant pressure” has a different meaning than “poor appetite” or “poor mental state”, where “poor” serves different purposes. And (2) ambiguous entity classification: In some cases, it is difficult to determine the exact entity type, leading to overlapping entity classes. Common symptom entities may also be disease or diagnostic entities. For example, “upper respiratory infection” can be both a symptom and a disease name.

### 2.2. Redefinition of Entity Types and Processing Strategy

The publicly available Yidu-S4K dataset was manually adjusted, with the focus on the accuracy of the recognition model. Some of the entity types have insufficient association with tumor prognosis. Therefore, it becomes relevant to redefine the entity types and reannotate the dataset.

Defining the entity type strategy: analyzing the existing electronic medical records of the same type of electronic medical records and clinical medical knowledge, furthermore, combining with the tumor prognosis analysis and prediction scenario, the information contained can be classified into the following five aspects:(1)Specific cancer types of patients;(2)Various examinations conducted during the treatment period;(3)Various symptoms exhibited by patients during the treatment period;(4)Various treatment methods received by patients during medical care;(5)The physical condition of patients after medical intervention.

Therefore, the textual information in electronic medical records could be classified into the following five types of entities: disease diagnosis, auxiliary examinations, symptom manifestations, treatment measures, and physical condition. The entity classification is shown in Table 2. According to the defined entity types, 1000 training samples from the original dataset can be reannotated. The data processing workflow is shown in the Figure 1.

A preprocessing strategy is designed for Chinese electronic medical texts, which encompasses the redefinition of entity types, the removal of outliers and invalid characters, and the supplementation of incomplete sentences. By analyzing the relevant medical proprietary terms involved in the patient’s treatment process, five entity types are redefined as: disease diagnose, auxiliary examination, symptom manifestation, treatment measure, and physical condition. Subsequently, invalid characters, such as spaces, newline characters, or garbled text from the dataset are eliminated, as they could potentially affect the model’s recognition accuracy. Following that, data desensitization is performed to exclude any sensitive information, such as patient names or medical identification numbers. Finally, we augmented sentences that were either incoherent or incomplete. For example, in cases where there is a lack of descriptive modifiers for a patient’s mental, dietary, or sleep conditions, it is modified to ’normal’ if the patient’s postoperative condition is favorable, or modified to ’average’ if the postoperative recovery is less favorable. Adhering to these strategies, we annotated the entity types in the data, performed validation checks, and finally generated an output text dataset.

After reannotating the entire training dataset, the resulting data are exported as BIO data files for subsequent entity recognition algorithms. An example of the data file is shown in Figure 2.

The data example in Figure 3 adopts the BIO annotation scheme, where “B” represents the first character of an entity, “I” represents the remaining characters of an entity, and “O” indicates that a character does not belong to any entity type. If a character in the text belongs to an entity, its label is appended with the corresponding entity type after “B” or “I”. After complete annotation, the count of each entity in the training set is shown in Table 3. In the training set, there are a total of 14,457 entities, including 2239 entities of the Disease Diagnosis type, 1531 entities of the Auxiliary Examinations type, 4650 entities of the Symptom Manifestations type, 1961 entities of the Treatment Measures type, and 1822 entities of the Physical Condition type.

## 3. Methodology

Inspired by the BERT and LSTM models, we propose a novel network model, which we call ALBERT-BiLSTM-MHA-CRF. The architecture of the proposed model is illustrated in Figure 4.

As can be seen from Figure 4, the proposed network mainly consists of four parts. The first part is ALBERT. In comparison with the BERT model, the ALBERT model consists of 12 layers of transformers, but it increases the hidden size, which refers to the number of features in the embeddings of each layer. The ALBERT pre-trained model is used to obtain the word vector sequence $X=\left(X_{1},X_{2},\ldots ,X_{n}\right)$ and vectorize the input sentences. The second part consists of two layers of the BiLSTM model, which takes the word vectors and learns contextual information to generate feature weights, resulting in a sequence of vectors with positional information. The third part is MHA. The sequence generated by the second part is input into the MHA. There are three different mapping operations involved in this process, namely transforming the input sequence into query, key, and value matrices. Subsequently, parallel self-attention operations are conducted on the sequence, resulting in representations, and the semantic information from all heads is continuously integrated to define it as the multi-head information. The multi-head information is ultimately mapped into an output matrix. The last part is CRF, which takes into account the order of output labels and optimizes the sequence by predicting labels based on their dependencies. This process aims to obtain the globally optimal sequence and generate the final output.

### 3.1. Enhanced Representation with Knowledge Fusion

In order to enhance the entity recognition task for electronic medical texts, an enhanced representation approach is introduced, which incorporates domain knowledge. A primary goal of this approach is to leverage domain-specific medical knowledge and integrate with text representation [30]. In this paper, the ALBERT pre-trained model is utilized to encode both the Chinese electronic medical text sentences *X* and the entity type descriptions *Q*, obtaining their respective token representations $h_{X}\epsilon R^{n\times d}$ and $h_{Q}\epsilon R^{\left|C\right|\times m\times d}$, where *n* and *m* represent the length of the electronic medical text and entity type description sentence, *d* is the vector dimension of the encoder, and $\left|C\right|$ is the number of entity types. The encoder’s computation formula is Equation (Equation 1):(1)$$h_{X}=f_{1}\left(X\right)\;\;\;h_{Q}=f_{1}\left(Q\right)$$

After obtaining the token $h_{Q}$ for type interpretation, the attention scores are calculated for each text representation $h_{xi}$ and each type description statement $h_{qi}$; then, we use the attention scores as weighting information to integrate the semantic meaning of type explanations into the tokens of the electronic medical text sentences. The attention score can be defined as follows:(2)$${a_{xi}}_{,q_{j}^{c}}=\frac{\exp\left(h_{xi}\cdot h_{q_{j}^{c}}\right)}{\sum_{j}\exp\left(h_{xi}\cdot h_{q_{j}^{c}}\right)}$$
(3)$$h_{x_{i}}^{c}=h_{x_{i}}+\sum_{j}a_{x_{i},q_{j}^{c}}\cdot h_{q_{j}^{c}}$$
(4)$${\tilde{h}}_{x_{i}}^{c}=\tanh\left(Vh_{x_{i}}^{c}+b\right)$$
where $x_{i}$ in Equation (Equation 2) is the i-th token of the text sentence *X*, 1⩽i⩽n; $h_{x_{i}}$ is the hidden vector generated for the i-th token; $q_{j}^{c}$ is the j-th token of the type description statement *Q*, where 1⩽j⩽m and cϵC; and $h_{q_{j}^{c}}$ is the hidden vector generated for the j-th token. A dot product operation between $h_{x_{i}}$ and $h_{q_{j}^{c}}$, followed by calculation of the correlation between $x_{i}$ and *c* by using Equation (Equation 3), results in $h_{x_{i}}^{c}$. Then, Equation (Equation 4) gives the hidden vectors ${\tilde{h}}_{x_{i}}^{c}$, where $V\epsilon R^{d\times d}$ and $b\epsilon R^{d}$ are learnable parameters of the network. These calculations are repeated for token $x_{i}$ and all type description statements, where vector ${\hat{h}}_{x_{i}}=\left({\hat{h}}_{x_{i}}^{1},\ldots ,{\hat{h}}_{x_{i}}^{\left|C\right|}\right)$ is obtained.

### 3.2. Multi-Head Attention Module for Polysemy Feature Extraction

The traditional BiLSTM model outputs hidden vectors with equal weights, which cannot fully capture the global information of the text sequence and the importance of each character within the sentence. For example, different words or characters in the same sentence often play different roles, and the same word or character may have significant variations in different sentences. Therefore, to assist rehabilitation robots in improving their understanding of words with different meanings in different contexts, the multi-head self-attention mechanism is introduced as a supplement. The MHA has several unique advantages [31,32]. Firstly, MHA captures the associative relationships between characters at any position in the sentence, enabling the model to learn long-range dependencies easily. Secondly, MHA produces output vectors using weighted sums, making the gradient propagation in the network model easier and reducing the likelihood of gradient exploding or vanishing. Moreover, MHA has strong parallel execution capabilities, leading to faster training speeds. By incorporating the MHA module, the proposed model can capture multiple semantic features at the character, word, and sentence levels.

For the vector matrix output from the BiLSTM model, the self-attention mechanism performs three different mapping operations to transform the vector matrix into three input matrices, each with the dimension of $d_{k}$: query *Q*, key *K* and value *V*. These matrices are then passed into the attention function, which calculates the weights on *V* based on the correlation between *Q* and *K*, resulting in corresponding vector representations. The calculation formula is defined as Equation (Equation 5):(5)$$A\left(Q,K,V\right)=softmax\left(\frac{QK^{T}}{\sqrt{d_{k}}}\right)V$$
where $Q\epsilon R^{m\times d_{k}}$, $K\epsilon R^{m\times d_{k}}$, $V\epsilon R^{m\times d_{k}}$, $d_{k}$ is the dimension of the hidden layer in the network, and $\sqrt{d_{k}}$ is the penalty factor for the inner product balance between *Q* and *K*.

In the multi-head self-attention mechanism, the query *Q*, key *K* and value *V* are linearly mapped independently *t* times using different parameter matrices. These mapped queries, keys and values are then fed into the respective *h* parallel heads to perform the attention function operation. This approach allows each parallel head to capture information about different representations of each character in the text sequence. Finally, the results from h parallel heads are combined, and undergo a linear transformation to obtain the final output. Equations (6) and (7) describe the computation process.
(6)$$head_{i}=attention\left(QW_{i}^{Q},KW_{i}^{K},VW_{i}^{V}\right)$$
(7)$$MH\left(Q,K,V\right)=concat\left(h_{1},h_{1},\ldots ,h_{h}\right)W^{o}$$
where $W_{i}^{Q}\epsilon R^{d_{k}\times d_{k}/h}$, $W_{i}^{K}\epsilon R^{d_{k}\times d_{k}/h}$, $W_{i}^{V}\epsilon R^{d_{k}\times d_{k}/h}$ and $W^{o}\epsilon R^{d_{k}\times d_{k}}$ represent the parameter matrices used for linear mapping, the $head_{i}$ denotes the i-th head in the multi-head self-attention mechanism, and the symbol concat represents the concatenation operation. MHA algorithm is elaborated in Algorithm 1.
**Algorithm 1** MHA algorithm**Require:** x-Input sequence (batch-size, seq-len, input-dim); mask (Mask for handling padding); num-heads (Number of attention heads).**Ensure:** best-path (Best path sequences).1:Initialize an empty list multihead-outputs for multi-head weights.2:For each attention head head in range(num-heads).3:   Generate query, key, and value matrices using linear transformations:4:      query = Linear(x); key = Linear(x); value = Linear(x);5:   Compute attention scores. scores = Dot-Product-Attention(query, key);6:   Calculate attention weights. attention-weights = Softmax(scores) (5);7:   Compute the output for each head:8:      head-output = Dot-Product-Attention(attention-weights, value) (6);9:   Add head-output to the multihead-outputs list;10:Concatenate and linearly transform the outputs from different heads in the multihead-outputs list (7)11:final-output = Linear(Concatenate(multihead-outputs))12:End

### 3.3. Conditional Random Field for Ambiguous Entity Classification

Although the LSTM model and MHA mechanisms learn contextual labels and output the most probable labels, they do not consider the dependencies between labels, which may result in the rehabilitation robots being unable to accurately recognize entity boundaries. The CRF model can take into account the order of labels. Therefore, the CRF model is chosen as the final output layer to handle ambiguous entity classification [33,34]. See Algorithm 2.
**Algorithm 2** CRF algorithm**Require:** H-Hidden states from BiLSTM (batch-size, seq-len, hidden-dim).**Ensure:** Best path sequences.1:Initialize the score matrix and best path matrix;2:For each sentence: batch in range(batch-size);3:   Calculate the effective length of the sentence:4:      seq-len = length of the current sentence;5:   Initialize the path probability matrix;6:   Initialize the best path matrix;7:   Compute the scores for the start labels:8:      viterbi[batch, 0] = scores[batch, 0] + transitions[START-TAG, :];9:   Perform forward pass;10:   Compute scores for the stop labels:11:      terminal-scores=viterbi[batch, seq-len-1, :] + transitions[:, STOP-TAG];12:   Find the best path’s stop label: best-tag = argmax(terminal-scores) (8);13:   Backtrack to retrieve the best path:14:      best-path[batch] = backtrace(backpointer[batch], best-tag);15:End

The linear chain CRF is defined as follows: a sequence of random variables $X=\left(X_{1},X_{2},\ldots ,X_{n}\right)$ and a corresponding sequence of random variables $Y=\left(Y_{1},Y_{2},\ldots ,Y_{n}\right)$ represent by linear chains, if the conditional probability distribution $P\left(Y|X\right)$ of the random variable sequence *Y*, given the sequence of random variables *X*, conforms to the formulation shown in Equation (Equation 8), then $P\left(Y|X\right)$ is referred to as a linear chain CRF. Therefore, NER tasks can be formulated as tagging problems, where *X* represents an input observation sequence, and *Y* represents a sequence of tags or states to be predicted. CRF algorithm is elaborated as Equation (Equation 8):(8)$$P\left(Y_{i}|X,Y_{1},\ldots ,Y_{i-1},Y_{i+1},\ldots ,Y_{n}\right)=P\left(Y_{i}|X,Y_{i-1},Y_{i+1}\right)$$

### 3.4. Loss Function

During the training process of the neural network model, it is essential to define the loss function to measure the error between predicted and true values, enabling the backward update of node weights. For the NER task, the model’s output can be regarded as a multi-classification task. Therefore, the cross-entropy loss function which is more suitable for measuring multi-classification tasks is chosen in this paper. Its calculation formula is shown as Equation (Equation 9), where *N* is the number of samples; *M* is the number of classes; $y_{ic}$ is the symbol function; if the true class of sample *i* is equal to *c*, the target value is set to 1, otherwise it is set to 0; and $p_{ic}$ represents the predicted probability of sample *i* belonging to class *c*.
(9)$$L=-\frac{1}{N}\sum_{i=1}^{N}\sum_{c=1}^{M}y_{ic}\log\left(p_{ic}\right)$$

The optimizer used in the model also has an impact on the training speed and accuracy to a certain extent. The Adam optimizer is selected as the model’s optimizer. The Adam optimizer combines the momentum-based gradient descent algorithm and the RMSProp algorithm, determining the next move direction based on the moving average of gradients and setting the step size based on a global upper bound. It has advantages such as simplicity, high computational efficiency, and low memory consumption.

### 3.5. Model Training and Analysis

Our model was trained and tested in Python 3.6 and Tensorflow 1.14 framework. The simulations were performed by using 3080Ti GPU hardware. The experiment employed Albert-based part with 12 multi-head attention mechanisms. The first hidden layer state of the BiLSTM part is set as 128 dimensions, and the second hidden layer state is set as 64 dimensions. The maximum sequence length is set as 512. The learning rate of the model is set as 2 × 10^−4^, and a dropout of 0.5 is applied to prevent overfitting. The batch size for training data are set as 16, and the proposed ALBERT-BiLSTM-MHA-CRF model is trained for a maximum of 50 iterations.

According to Algorithm 3, after defining the neural network structure and initializing the neural connections’ weights based on the input network parameters, training of the ALBERT-BiLSTM-MHA-CRF model starts. During each iteration, the cross-entropy loss between the neural network’s output and the true values is calculated. The gradients are then computed based on the learning rate, and the weights of individual neurons are updated accordingly. During the experiment, the changes in the loss function and accuracy were recorded for each training epoch. The resulting curves are shown in Figure 5. From the figure, it can be seen that during the early iterations, the model’s loss function value rapidly decreases, and the accuracy increases quickly. After approximately 20 iterations, the model reaches a relatively stable state, with subsequent fluctuations remaining steady. The low value of the loss function and the high accuracy indicate that the model performs well.
**Algorithm 3** Training of the ALBERT-BiLSTM-MHA-CRF**Require:** The maximum text length, dropout rate, the number of nodes in the first BiLSTM layer, the number of nodes in the second BiLSTM layer, the number of attention heads in the multi-head self-attention mechanism, batch size, learning rate and the number of iterations.**Ensure:** The optimal precision, recall, and F1-Score of the model under the specified number of experimental runs.1:Initialize the connection weights of neurons within (0, 1);2:for i = 1 … epochs do:3:   Calculate the output of the ALBERT-BiLSTM-MHA-CRF model based on the current sample data;4:   Calculate the cross-entropy loss between the output of the neural network and the ground truth;5:   Compute the gradient of each weight for descent;6:   Update the weights based on the descent gradient;7:   Calculate the loss function value (9), accuracy, precision, recall and F1-Score after each iteration;8:End

## 4. Experimental Analysis and Discussion

### 4.1. Evaluation Metrics

The averages are employed as the evaluation metrics for the named entity recognition model. Evaluation of the named entity recognition model is based on the following criteria:(1)Whether the entity type is labeled correctly;(2)Whether the boundary of the entity is identified correctly.

The output is considered correct only if both the resulting entity type and entity boundary are correct. If only the entity type is correct but the entity boundary is not, or if the entity boundary is correct but the entity class is not recognized, it is considered incorrect. The calculated values and implications of the formula for calculating accuracy, recall, and F1-Score are presented in Table 4.

Precision represents the proportion of true positive instances among the instances predicted as positive by the model, and it is defined as follows:(10)$$Precision=\frac{TP}{TP+FP}$$

In terms of the named entity recognition the sensitivity of the algorithm is represented as the fraction of correctly recognized entities among all the retrieved instances, i.e.,
(11)$$Recall=\frac{TP}{TP+FN}$$

Precision and recall may not be compatible in certain situations, leading to cases where precision is high but recall is low, or vice versa. To comprehensively reflect the results of precision and recall, the F1-Score is commonly used. It is the harmonic mean of precision and recall, defined as follows:(12)$$F1=\frac{2\cdot Precision\cdot Recall}{Precision+Recall}$$

In addition to calculating precision, recall and F1-Score for each individual class, it is also necessary to use weighted averaging to obtain the average performance across multiple classes. The calculation of weighted averaging is shown as follows:(13)$$Weighted\left(Z\right)=\sum_{i=1}^{N}x_{i}Z_{i}$$
where $Z_{i}$ represents an evaluation metric for individual class, *N* denotes the total number of classes, and $x_{i}$ indicates the class weight.

### 4.2. Ablation Experiments and Analysis

To verify the effectiveness of each module of the model, ablation experiments were designed to compare the model with the ALBERT-BiLSTM, ALBERT-BiLSTM-MHA, and ALBERT-BiLSTM-CRF separately, aiming to validate the performance of the MHA and the CRF module.

During the experimental process, the change curves of the four models’ loss functions and accuracies at each training epoch are recorded, as shown in Figure 6.

As a result of the variation graphs, it may be inferred that the proposed ALBERT-BiLSTM-MHA-CRF network has advantages over the other three models in terms of loss function and accuracy. Note that all four models show a rapid decrease in the loss function and significant increase in the accuracy for relatively low values of epoch numbers. As the epoch number increases, the ALBERT-BiLSTM-MHA-CRF model exhibits lower loss function values and higher accuracy in comparison with other models. This indicates that the number of iterations in BiLSTM and MHA modules play a crucial role in performance of the model.

In order to calculate the precision, recall, and F1-Score of five entity types that output from the four models, 10 experiments are conducted, and the average results are shown in Table 5 and Figure 7.

The following conclusions can be drawn from Table 5 and Figure 7:(1)Comparing ALBERT-BiLSTM-MHA-CRF with ALBERT-BiLSTM-CRF, the former demonstrates superior performance in recognizing entities related to disease diagnosis, symptom manifestations, and physical condition. MHA captures contextual information, thereby enhancing entity recognition accuracy.(2)ALBERT-BiLSTM-MHA-CRF outperforms ALBERT-BiLSTM-MHA across all five entity types. This suggests that the CRF model can better achieve precise boundary delineation, improving the model’s accuracy in entity recognition.(3)The model exhibits relatively poorer recognition performance for the “treatment measures” entity type. The diversity and complexity of treatment measures, coupled with the non-standardized descriptions of these measures in medical texts, make their recognition more challenging.

The weights for the five entity types are as shown in Table 6. To analyze the overall effects of the ablation experiments on the models, the weighted average of the four models is summarized and presented in Figure 8. From the figure, it can be observed that the proposed ALBERT-BiLSTM-MHA-CRF model outperforms the ALBERT-BiLSTM-CRF model with a precision improvement of 0.01, a recall improvement of 0.02, and an F1-Score improvement of 0.01. This indicates that the inclusion of the MHA module allows the model to capture contextual semantic features from multiple dimensions, thereby enhancing the recognition accuracy. Similarly, the proposed ALBERT-BiLSTM-MHA-CRF model performs better than the ALBERT-BiLSTM-MHA model, with a precision improvement of 0.06, a recall improvement of 0.07, and an F1-Score improvement of 0.04. This indicates that the addition of the CRF module effectively utilizes the contextual information in the text, leading to improved model performance.

### 4.3. Comparison Experiment and Analysis

To further validate the effectiveness of the model, a comparative experiment is designed to compare the model with BERT-BiLSTM-CRF [35], BERT-Softmax [36], BiLSTM-CRF [37], and BiLSTM-softmax [38]. The loss and accuracy variation curves for the five models at each training epoch are shown in Figure 9.

From Figure 9, it can be observed that the ALBERT-BiLSTM-MHA-CRF model demonstrates a significant advantage in both the loss function and accuracy. In the early stages of training, all five models exhibit a rapid decrease in the loss function and a quick increase in accuracy. Among the models, BiLSTM-CRF and BiLSTM-Softmax, which do not incorporate pre-trained models, show the fastest variations. The models that utilize ALBERT come next, followed by BERT which has the most parameters. Consequently, BERT-BiLSTM-CRF and BERT-Softmax exhibit slower variations. However, as the training progresses, the proposed ALBERT-BiLSTM-MHA-CRF model achieves the lowest loss function value and the highest accuracy, confirming its superiority over the other models.

The precision, recall and F1-Score of five entity types that output from the five comparison models are calculated separately. The average results obtained from 10 experiments and the weighted average values of the five models in the comparative experiments are presented in Table 7.

Based on the comparative experimental results from Table 7 and Figure 10, the ALBERT + BILSTM + MHA + CRF model demonstrates significant advantages in named entity recognition tasks. Compared to other models, such as the BiLSTM + Softmax model, this model shows notable improvements in P, Recall, and F1-Score. Specifically, for the key entity types “Disease Diagnosis”, “Symptom Manifestation”, and “Treatment Measures”, the F1-Score of this model are increased by over 0.10. These improvements are attributed to the strong representation capabilities of the ALBERT pre-trained model, the context modeling ability of MHA, and the precise boundary prediction provided by CRF, which effectively addresses issues related to varying meanings of the same entity in different contexts and fuzzy entity boundaries. However, it is noteworthy that, despite the overall superior performance of the model, the recognition performance for the “Treatment Measures” category remains suboptimal, with an F1-Score of only 0.83. This indicates that challenges persist in processing entities of this category, and further research is needed to enhance the accuracy of recognition for this type of entity, thereby improving the model’s performance across all entity categories.

Although the BERT-BiLSTM-CRF model excels in named entity recognition, it faces significant computational challenges when dealing with Chinese electronic medical records. While BERT provides deep semantic representation and rich contextual information, its large number of parameters (BERT 110M, BiLSTM 6M, CRF 25) results in substantial computational overhead. In contrast, ALBERT significantly reduces the number of parameters to approximately 60M through parameter sharing and introduces an additional 6M parameters for the MHA. This results in the total number of parameters for BERT-BiLSTM-CRF being about 1.61 times that of the ALBERT-BiLSTM-MHA-CRF model. This optimization markedly improves the training efficiency of the model. For each training epoch, the BERT-BiLSTM-CRF model requires approximately 5 min, while the ALBERT-BiLSTM-MHA-CRF model requires only about 3.5 min, demonstrating a clear advantage in training time. Consequently, the ALBERT-BiLSTM-MHA-CRF model not only enhances accuracy, but also optimizes computational efficiency and resource utilization, showing superior practicality and performance for complex named entity recognition tasks.

During the modeling process, the results of each model run may vary due to randomness. To compare the stability of different models, the precision, recall, and F1-Score of the five models are calculated for each fold in a 10-fold cross-validation. Box plots were then generated, as shown in Figure 11, to visualize the distribution of these metrics across the folds. The upper and lower whiskers represent the range of the data, the height of the box reflects the dispersion of the data, with a shorter box indicating that the data is concentrated and has low variance, and the median line indicates the median of the metric. According to the graph, it can be observed that the precision, recall, and F1-Score of the proposed ALBERT-BiLSTM-MHA-CRF model in the 10 test sets are all within the upper and lower limit ranges, with no outliers, and the data distribution is relatively concentrated. This indicates that there is not much variation in the results during the 10-fold cross-validation, and the proposed ALBERT-BiLSTM-MHA-CRF model is relatively stable, making it suitable for predictive analysis in Chinese electronic medical records.

Based on the experimental results, the algorithm developed in this study demonstrates significant advantages in accuracy and practical application. The model substantially improves entity recognition accuracy, reducing common issues of misidentification and missed identifications present in traditional methods. This is particularly crucial for handling complex medical records, as it ensures that the robot can accurately extract key information and avoid potential errors in medical decision-making. Additionally, the improved algorithm enhances the robot’s ability to understand complex medical language, effectively addressing issues of terminology diversity and contextual differences, thus optimizing the overall performance of the system. This method offers significant advantages for rehabilitation robots. Firstly, the precise entity recognition capability allows robots to understand and process medical records more effectively, thereby enhancing the accuracy and efficiency of medical decision-making. Secondly, the improved recognition ability enables robots to handle more complex medical information, reducing the risk of medical errors due to misidentification. Additionally, the advancement of this technology promotes the application of rehabilitation robots in medical environments, allowing them to provide more reliable decision support, which improves treatment outcomes and the quality of medical services. These advancements not only enhance the functionality of rehabilitation robots, but also drive further development in the medical field.

## 5. Conclusions

To enable medical rehabilitation robots to effectively recognize named entities in Chinese electronic medical texts and improve the quality and efficiency of medical services, the effectiveness of the Albert-BILSTM-MHA-CRF model is evaluated in this study. An innovative medical entity recognition method is proposed in the experiment. By integrating data preprocessing strategies with a knowledge-fusion medical entity recognition model, challenges posed by dense domain knowledge in knowledge fusion representation are successfully addressed, along with the issue of diverse categorizations of the same entity in different contexts. This integration also effectively handles the blurred entity boundaries caused by the cross-expression of different entities. Experimental results fully demonstrate the significant effectiveness and superiority of the model in the task, providing a viable solution for named entity recognition in Chinese electronic medical records used by medical rehabilitation robots. Future research is expected to focus on the following aspects: Firstly, further optimization of the model structure will be prioritized, with an emphasis on incorporating more complex attention mechanisms or updated pre-trained language models. This is aimed at improving the accuracy and efficiency of named entity recognition, thereby providing more precise and efficient solutions in the field of medical rehabilitation. Secondly, the applicability of this model to other domains, such as legal texts and financial reports, which are highly specialized and complex, will be explored to assess the model’s generalization capabilities across different fields.

## Figures and Tables

**Figure 1 sensors-24-05624-f001:**
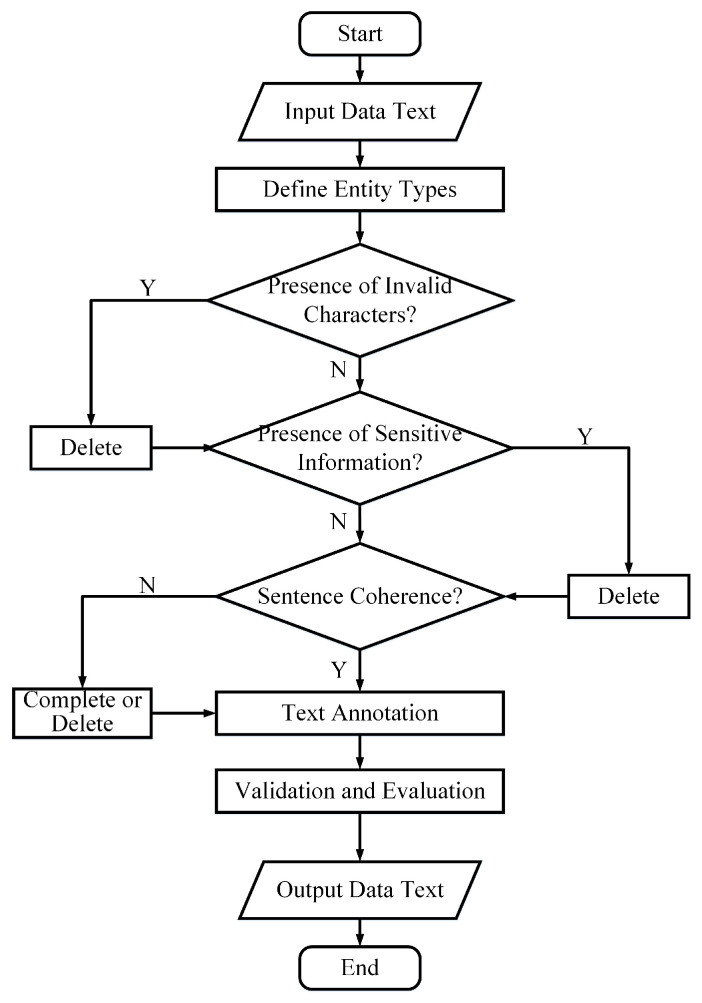
Flowchart of data preprocessing.

**Figure 2 sensors-24-05624-f002:**
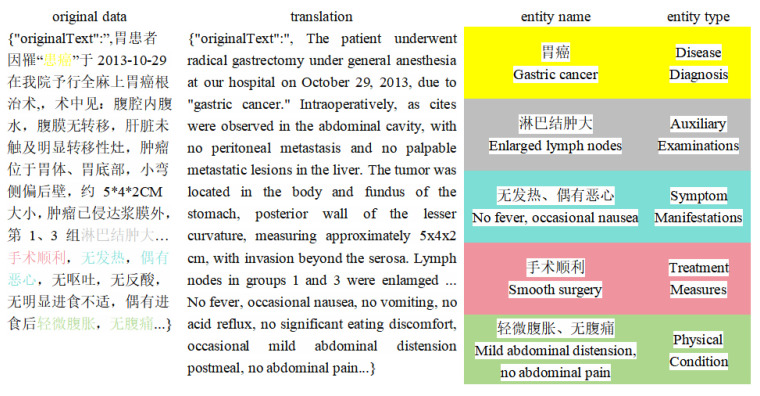
Result of data processing.

**Figure 3 sensors-24-05624-f003:**
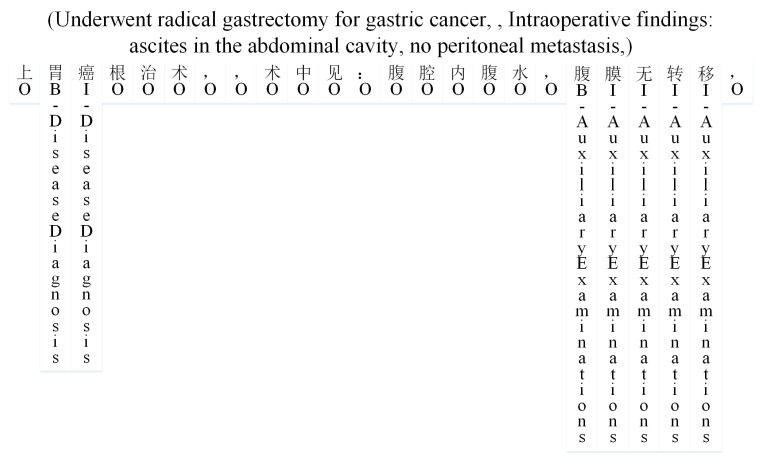
Result of data processing.

**Figure 4 sensors-24-05624-f004:**
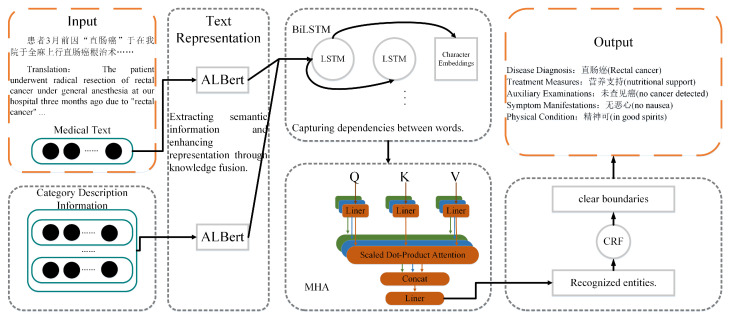
ALBERT-BiLSTM-MHA-CRF framework diagram.

**Figure 5 sensors-24-05624-f005:**
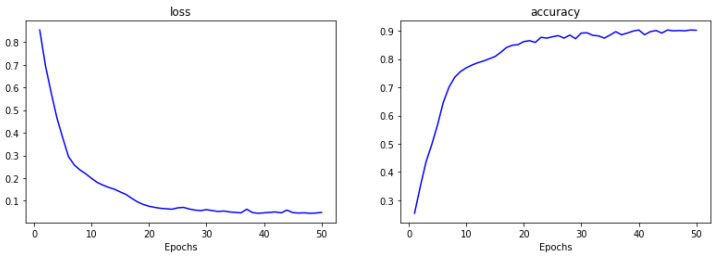
ALBERT-BiLSTM-MHA-CRF model loss and accuracy variation curve.

**Figure 6 sensors-24-05624-f006:**
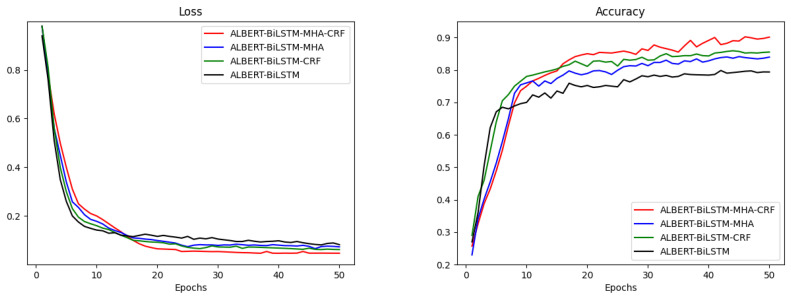
Loss and accuracy of ablation experiments variation curve.

**Figure 7 sensors-24-05624-f007:**
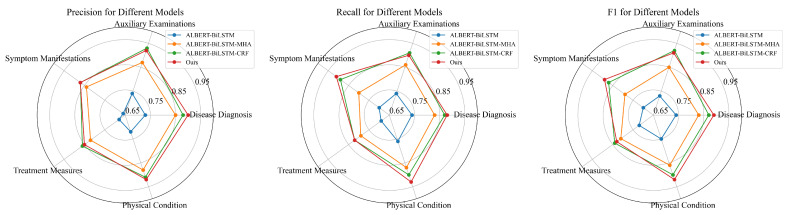
Ablation experiment results.

**Figure 8 sensors-24-05624-f008:**
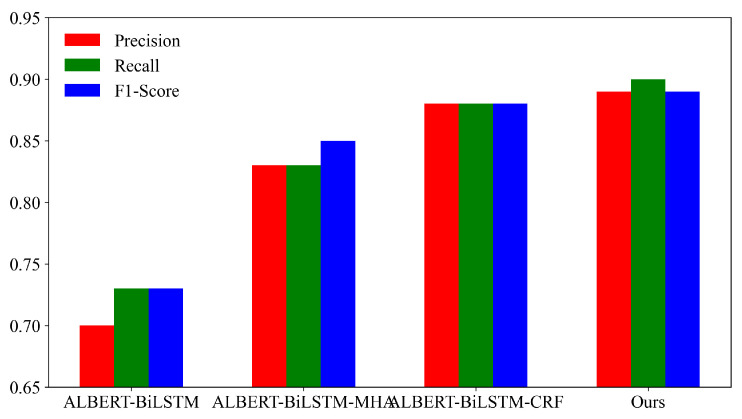
Weighted average results of ablation experiments.

**Figure 9 sensors-24-05624-f009:**
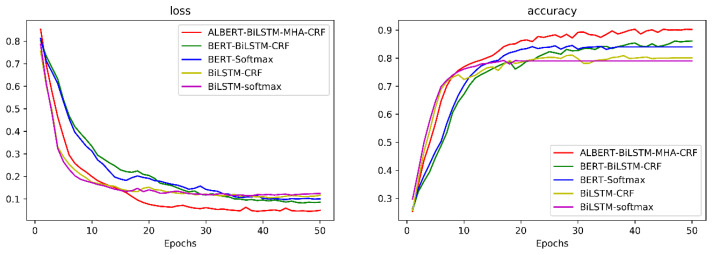
Loss and accuracy of comparative experiments variation curve.

**Figure 10 sensors-24-05624-f010:**
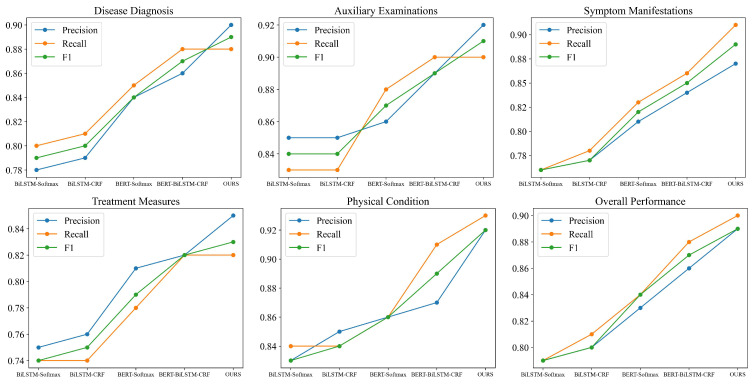
P, R, and F1 of five types in comparative experiments.

**Figure 11 sensors-24-05624-f011:**
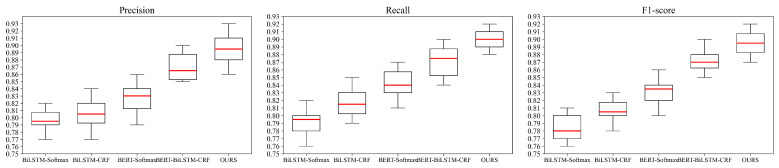
Cross validation of box plot by comparison experiment.

**Table 1 sensors-24-05624-t001:** Number of entities per class in the original dataset.

Entity Type	Disease Diagnosis	Inspection	Check	Surgery	Medicine	Anatomical Site
train	4208	963	1179	1029	1822	8417
test	1322	348	585	162	485	3093

**Table 2 sensors-24-05624-t002:** Electronic medical record entity classification.

Entity Type	Type Definition	Examples
Disease	Specific cancer types of patients	Gastric cancer, Colon
Diagnosis		cancer, Liver cancer
Auxiliary	Cancer-related examinations conducted	Cancer metastasis,
Examinations	during the treatment period	Cancer embolism
Symptom	Discomfort and manifestations exhibited	Fever, Nausea,
Manifestations	by patients during treatment	Vomiting
Treatment	Intervention measures taken for	Acid suppression,
Measures	patient’s disease	Antiemetics
Physical	Patient’s physical condition after	Mental state, Dietary
Condition	medical intervention	status, Sleep quality

**Table 3 sensors-24-05624-t003:** Number of entities of each type.

Entity Type	Quantity
Disease Diagnosis	2239
Auxiliary Examinations	1531
Symptom Manifestations	4650
Treatment Measures	1961
Physical Condition	4076

**Table 4 sensors-24-05624-t004:** Indicator types.

	Recognition Is Positive	Recognition Is Negative
Reality is positive	TP	FN
Reality is negative	FP	TN

**Table 5 sensors-24-05624-t005:** Comparison of model ablation experiment results.

	Evaluation Metrics	Disease Diagnosis	Auxiliary Examinations	Symptom Manifestations	Treatment Measures	Physical Condition
ALBERT-BiLSTM	P	0.73	0.74	0.66	0.68	0.72
	Recall	0.74	0.74	0.70	0.69	0.76
	F1	0.74	0.73	0.70	0.72	0.75
ALBERT-BiLSTM-MHA	P	0.85	0.87	0.84	0.82	0.88
	Recall	0.83	0.86	0.80	0.79	0.87
	F1	0.83	0.85	0.79	0.81	0.86
ALBERT-BiLSTM-CRF	P	0.88	**0.93**	**0.87**	**0.86**	0.91
	Recall	0.87	**0.91**	0.89	**0.82**	0.90
	F1	0.87	**0.92**	0.87	**0.84**	0.90
Ours	P	**0.90**	0.92	**0.87**	0.85	**0.92**
	Recall	**0.88**	0.90	**0.91**	**0.82**	**0.93**
	F1	**0.89**	0.91	**0.89**	0.83	**0.92**

**Table 6 sensors-24-05624-t006:** Five types of entity weights.

Entity Type	Quantity	Weights
Disease Diagnosis	2239	0.155
Auxiliary Examinations	1531	0.105
Symptom Manifestations	4650	0.322
Treatment Measures	1961	0.136
Physical Condition	4076	0.282

**Table 7 sensors-24-05624-t007:** The results of the comparative experimental models.

	Evaluation Metrics	Disease Diagnosis	Auxiliary Examinations	Symptom Manifestations	Treatment Measures	Physical Condition	Overall Performance
BiLSTM-Softmax	P	0.78	0.85	0.76	0.75	0.83	0.79
	Recall	0.80	0.83	0.76	0.74	0.84	0.79
	F1	0.79	0.84	0.76	0.74	0.83	0.79
BiLSTM-CRF	P	0.79	0.85	0.77	0.76	0.85	0.80
	Recall	0.81	0.83	0.78	0.74	0.84	0.81
	F1	0.80	0.84	0.77	0.75	0.84	0.80
BERT-Softmax	P	0.84	0.86	0.81	0.81	0.86	0.83
	Recall	0.85	0.88	0.83	0.78	0.86	0.84
	F1	0.84	0.87	0.82	0.79	0.86	0.84
BERT-BiLSTM-CRF	P	0.86	0.89	0.84	0.82	0.88	0.86
	Recall	**0.88**	**0.90**	0.86	**0.82**	0.91	0.88
	F1	0.87	0.89	0.85	0.82	0.89	0.87
Ours	P	**0.90**	**0.92**	**0.87**	**0.85**	**0.92**	**0.89**
	Recall	**0.88**	**0.90**	**0.91**	**0.82**	**0.93**	**0.90**
	F1	**0.89**	**0.91**	**0.89**	**0.83**	**0.92**	**0.89**

## Data Availability

The data used in the study is sourced from CCKS2019 Task 1: Medical Entity Recognition and Attribute Extraction for Chinese Electronic Medical Records. This dataset is specifically designed for researching named entity recognition in electronic medical records, with high usability. The dataset’s website link is https://github.com/Bureaux-Tao/ccksyidu4k-ner (accessed on 15 August 2023).

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
