# Peer review of "Recognition of Chinese Electronic Medical Records for Rehabilitation Robots: Information Fusion Classification Strategy"

_sensors, 2024, doi:10.3390/s24175624_

Round 1

Reviewer 1 Report

Comments and Suggestions for Authors

In this paper, a medical entity recognition algorithm, which is based on a fusion classification strategy, is developed to enhance the processing and understanding capabilities of rehabilitation robots for patient data. Specifically, a preprocessing strategy is proposed according to clinical medical knowledge, which includes redefining entities, removing outliers, and eliminating invalid characters. Subsequently, a medical entity recognition model is developed to identify Chinese electronic medical records, thereby enhancing the data analysis capabilities of rehabilitation robots. To extract semantic information, the ALBERT network is utilized, and BILSTM and MHA networks are combined to capture the dependency relationships between words, overcoming the problem of different meanings for the same entity in different contexts. The CRF network is employed to determine the boundaries of different entities. The research results indicate that the proposed model significantly improves the recognition accuracy of electronic medical texts by rehabilitation robots, demonstrating high practical utility. Generally speaking, the topic of this paper is very interesting and valuable. My comments are as follows:

(1) Please explain more details about your contributions so that the readers of the paper can understand it.

(2) Some of the references in the introduction can be updated, so it is suggested that they be replaced with references from the last three to five years.

(3) Some remarks or explanations could be added to explain the advantage of the proposed method, especially in the field of robot applications.

(4) In Table 5, Table 7 and Table 8, the authors are advised to highlight the best results for each indicator.

(5) Some Figures in this manuscript are of low quality, for example Figure 1 and Figure 2. I hope the authors pay attention to colormaps and resolution to meet the publishing requirements.

(6) There are a few minor editorial mistakes throughout the paper. Please carefully check the paper and improve the organizations.

(7) The authors are advised to re-edit the Reference to ensure the references’ format is consistent with each other, for example [2],[7]. 

(8) Some further research topics could be discussed in the conclusion.

 Overall, this paper should be re-reviewed provided that the above comments are carefully taken into account.

Reviewer 2 Report

Comments and Suggestions for Authors

The main aim of the paper is to introduce a medical entity recognition method for Chinese electronic medical records that enhances the processing and understanding properties of rehabilitation robots for patient data. The proposed method uses a preprocessing stage consisting of redefining entities, removing outliers and invalid characters, and supplementing incomplete sentences. The main stage, the recognition step, is carried out by a neural network-based algorithm that combines ALBERT, two-layer LSTM, MHA, and CRF respectively.

The performance of the proposed model is measured in terms of precision, recall, and F1 score. The provided results suggest that the proposed model is quite accurate.

The paper is well-written, the methodology is clearly explained, and the results look promising. The following suggestions should be taken into account:

1.     The importance of the study together with the original elements of the reported work should be highlighted at the end of the introductory part.

2.     The statistical analysis reported in Section 4.3 should be extended to more experiments (say 50-100).

3.     The analysis of the complexity of the resulting algorithm against BERT-BiLSTM-CRF should be provided (the most accurate existing alternative according to the authors). 

Comments on the Quality of English Language

Minor revision
